evolution/taxonomy and systematics/ecology

biogeography, commensal, invasive species, Squamata, trans-Atlantic dispersal

**Author for correspondence:**
Ishan Agarwal
e-mail: ishan.agarwal@gmail.com

†Deceased.

# How the African house gecko (*Hemidactylus mabouia*) conquered the world

Ishan Agarwal[1,2], Luis M. P. Ceríaco[1,3],
Margarita Metallinou[1,†], Todd R. Jackman[1] and
Aaron M. Bauer[1]

[1]Department of Biology and Center for Biodiversity and Ecosystem Stewardship, Villanova University, 800 Lancaster Avenue, Villanova, PA 19085, USA
[2]Thackeray Wildlife Foundation, Vaibhav Chambers, Bandra, Mumbai 400051, India
[3]Museu de História Natural e da Ciência da Universidade do Porto, Praça Gomes Teixeira, 4099-002 Porto, Portugal

IA, 0000-0001-9734-5379; LMPC, 0000-0002-0591-9978;
TRJ, 0000-0002-5391-8136; AMB, 0000-0001-6839-8025

Alien species are among the greatest threats to biodiversity, but the evolutionary origins of invasiveness remain obscure. We conducted the first range-wide sampling of *Hemidactylus mabouia* from more than 120 localities across Africa, Madagascar and the Neotropics to understand the evolutionary history of one of the most widely distributed, invasive vertebrates in the world. We used a multi-locus phylogeny, species delimitation, fossil-calibrated timetree, ancestral area reconstruction and species distribution models (SDMs) to determine how many putative species-level lineages are contained within *H. mabouia*, the timing and tempo of diversification, and the origins of commensality—providing insights into the evolutionary origins of invasiveness. Our analyses suggest '*H. mabouia*' originated in the Miocene in the Zambezian biogeographic region and includes as many as 20 putative species-level lineages, of which only *Hemidactylus mabouia sensu stricto* is invasive and widely distributed, including all Neotropical records. Zambezia is the hotspot for diversity within the group with 14 species in southeastern Zambezia. SDMs suggest that *H. mabouia* was able to establish in the Neotropics due to habitat suitability, and globalization and the slave trade probably allowed it to cross the Atlantic. Distribution models for the *H. mabouia* complex overpredict the range of the invasive *H. mabouia sensu stricto*—highlighting the importance of taxonomy in invasive species management.

## 1. Introduction

Alien species are among the most prevalent threats associated with vertebrate extinctions in the Anthropocene [1]. Geckos

include some of the most invasive species of reptiles, and in some cases, there are known or predicted negative consequences of the invasives on native geckos (e.g. [2,3]; but see [4]). The cosmopolitan genus *Hemidactylus* stands out among geckos—while most of the more than 165 species have small ranges, approximately 10 species have achieved intercontinental distributions [5–7]. While the natural biogeographic history of the group has involved numerous overwater intercontinental dispersals at both recent and deep timescales [8,9], the globally distributed species are human commensals that have established across many parts of the world in the recent past in association with mass transportation of goods and people (e.g. the spread of *H. frenatus* in the Pacific in association with troop and supply movements during World War II; [6,10–12]; and the recent rapid spread of *H. turcicus* in the United States, e.g. [13,14]). Older, yet still anthropogenically mediated historical movement of *Hemidactylus* spp. has also been implicated; *Hemidactylus flaviviridis* was established in North Africa at least two centuries ago [15] but has an Indian origin [8,16] and *H. robustus* and other congeners have been proposed to have expanded along ancient Middle Eastern caravan routes [17,18].

As is the case with many widely distributed species, phylogenetic data reveals that some widespread invasive 'species' include multiple divergent species-level lineages, only a few of which may actually be invasive. For example, *H. brookii* was considered to have a near pan-tropical distribution [5], but African '*H. brookii*' (= *H. angulatus* complex) have been shown to represent a deeply divergent lineage within *Hemidactylus* with no close affinities to the clade containing *H. brookii sensu stricto* [19]. Even Asian *H. brookii* have been shown to be a complex of at least eight species, which includes probably only two species that are invasive outside of South Asia: *H. murrayi* in Southeast Asia and *H. parvimaculatus* in the islands of the Indian Ocean and scattered localities around the world [20–23]. *Hemidactylus* provides a useful system to understand what makes a species invasive, as there are multiple clades of invasive species that are sister to narrowly distributed species [8,21].

One of the most successful invaders among *Hemidactylus* is the type species of the genus, *H. mabouia*. This medium-sized gecko (snout to vent length less than 68 mm) is widely distributed in the Neotropics and sub-Saharan Africa (figure 1), with a presumed African origin [5,19]. The species was established in the Neotropics by at least the middle seventeenth century [24], and was described based on West Indian populations [5,25]. The species seems able to outcompete native species (e.g. [26]); and where it has become introduced in recent decades appears to have pushed out or out-competed earlier colonizers, as in Florida, where *H. turcicus* (established since approx. 1910) and the more recent colonizers *H. frenatus* and *H. garnotii* have been marginalized or replaced when *H. mabouia* arrived [27,28]. It has likewise displaced both native geckos and previously established invasives elsewhere in the Neotropics [29–31]. In addition to colonizing the Americas, *H. mabouia* has also spread within the last 40 years into temperate Africa, extending southward from subtropical KwaZulu-Natal into the Eastern Cape Province of South Africa [32–34].

The limited published mitochondrial sequence data for *H. mabouia* show conflicting signals, with almost no divergence from samples across Africa and the Neotropics [19] and cryptic diversity along the East African coast [35]. This suggests *H. mabouia* includes both widely distributed commensal lineages and others restricted to more modest areas of endemism; though the origin, number and distribution of commensal and potentially endemic lineages is unknown. We undertook the first range-wide sampling of *H. mabouia sensu lato* including more than 180 individuals from more than 120 localities in Africa and the Americas. We use a multi-locus phylogeny, tree-based species delimitation, a fossil-calibrated timetree, ancestral area reconstructions and species distribution modelling to estimate how many species-level lineages are contained within *H. mabouia*, the timing and tempo of diversification in Africa and the New World, and the number of independent origins of invasiveness within this group.

# 2. Methods

## 2.1. Taxon and gene sampling

We collated approximately 200 tissue samples from museums across the world and samples collected by our team and collaborators. We used Kluge's [5] definition of *Hemidactylus mabouia sensu stricto* (*ss*) on a sample of specimens to confirm which lineages matched the nominal species in morphological and meristic data (detailed morphological and taxonomic work in preparation; LMP Ceríaco, I Agarwal and AM Bauer, in preparation), and included other African *Hemidactylus* allied to the African-Atlantic clade [19,36]. The main dataset included up to four genes (up to 3274 aligned nucleotides, nt; mtDNA: ND2; nucDNA: MXRA5, PDC, RAG1) for 186 specimens of the *H. mabouia* group from 118

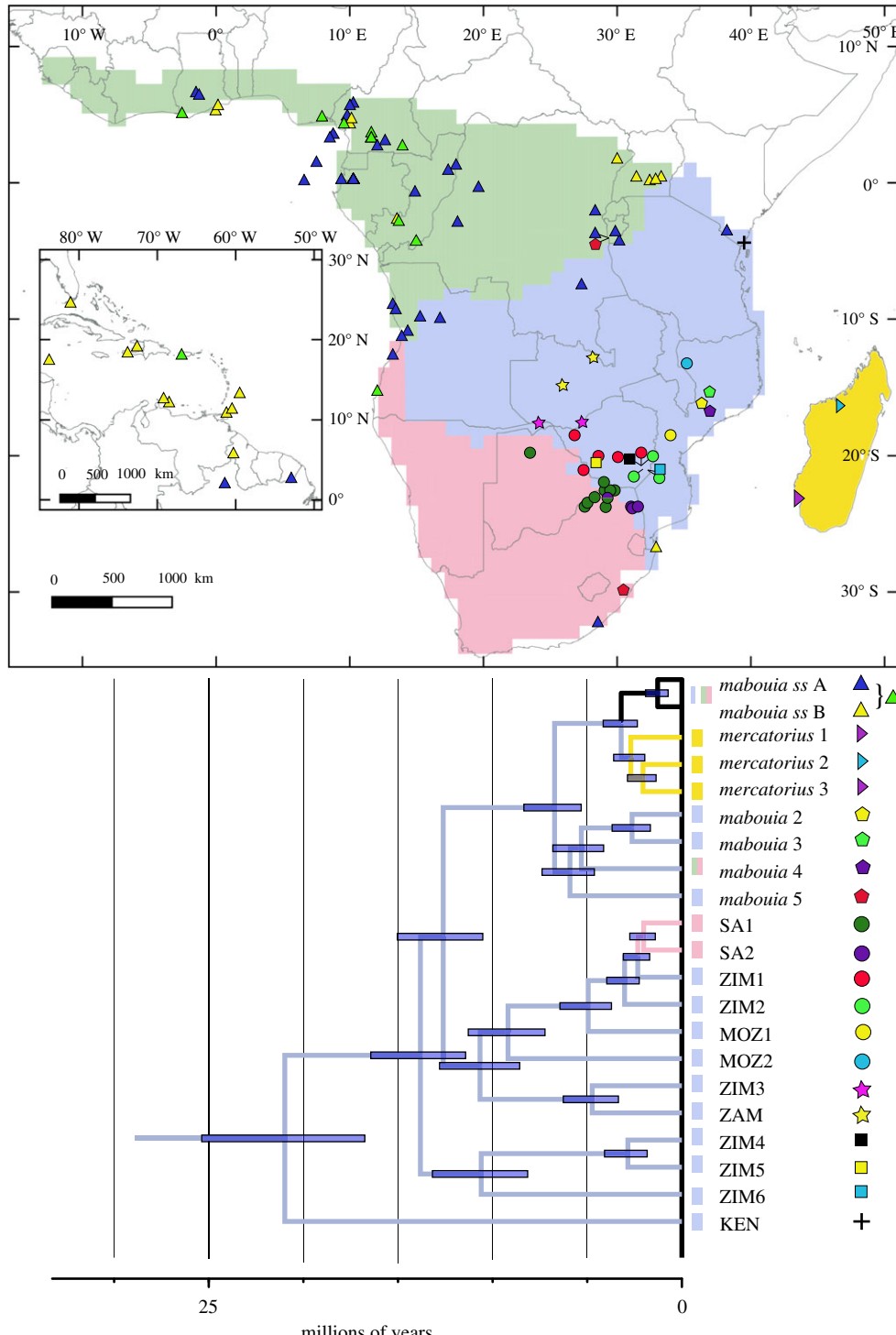

**Figure 1.** Top panel, sampled localities for the *Hemidactylus mabouia* complex from Africa and the New World (inset) with ancestral regions coloured: green = Congolian, blue = Zambezian, pink = Southern African, yellow = Madagascar, white (inset) = New World. Lower panel, timetree based on concatenated mitochondrial and nuclear data for the *H. mabouia* complex with most likely ancestral areas mapped onto branches (black = distributed in all areas except Madagascar) and 95% highest posterior density indicated by error bars, current distributions shown to the left of taxon labels and map symbols shown to the right. *H. mabouia ss* A and B are the only widely distributed, invasive lineages of the complex.

localities which were combined with published sequences of African *Hemidactylus* and the major global clades of the genus (electronic supplementary material, table S1). We also added sequences for the *H. platycephalus* group, which has historically been confused with *H. mabouia* [37]. DNA extractions were carried out using the Qiagen DNeasy kit and we used published primers (electronic supplementary

material, table S2) and PCR protocols to amplify target regions. Purification of PCR products used an in-house magnetic bead clean-up, and sequencing was outsourced to GENEWIZ (Plainfield, NJ, USA).

## 2.2. Phylogenetic analyses and species delimitation

We used ClustalW for sequence alignment as implemented in MEGA 5.2 [38] and translated protein-coding genes to check for premature stop codons. The best-fit partitioning scheme and models of sequence evolution were selected in PartitionFinder 2.1.1 [39] for ND2 (1041 nt) and the concatenated nuclear data (2263 nt; electronic supplementary material, table S3) using the Bayesian information criteria. Relationships were reconstructed using maximum likelihood (ML) using RAxML-HPC2 v. 8.2.12 [40] on XSEDE as implemented on the CIPRES web server (https://www.phylo.org/) with default settings and 1000 rapid bootstraps used to assess node support. Uncorrected genetic distance (%) was calculated in MEGA 5.2.

Tree-based automated species delimitation methods based on within and between species branching patterns can be used to identify candidate species. We used bPTP [41] for species delimitation on two independent datasets, the ND2 and the concatenated nuclear ML tree, implemented on the web server (http://species.h-its.org/ptp/), for 500 000 generations with a burn-in of 25%, thinning set to 100 and with the outgroups dropped. While more sophisticated multi-loci coalescent methods are now available for species delimitation, our dataset was largest for ND2 and the nuclear data were used to corroborate preliminary diversity flagged by the mtNDA. We also used the threshold of 5% uncorrected ND2 sequence divergence to identify candidate species, following previous studies of gekkonid geckos [22,42].

To explore patterns of phylogeography, we truncated an ND2 alignment for haplotype estimation, keeping only the widely distributed *H. mabouia ss* (see Results), also deleting the first 149 nucleotides of the alignment as well as sequences with missing data that were represented by complete sequences from the same locality (and were identical for at least 400 nt). Popart 1.7 [43] was used to construct haplotype networks using median joining [44] and TCS [45] with default settings.

## 2.3. Divergence dating and ancestral area reconstructions

The alignment for divergence dating analyses included a single representative of each putative species from the species delimitation analyses, besides a broader sampling of *Hemidactylus*, Gekkonidae and the Gekkota (electronic supplementary material, table S4) and included up to 3316 aligned nt from four genes (ND2, MXRA5, PDC, RAG1). The analyses were conducted in BEAST 1.10.4 [46] using the partitions specified by PartitionFinder2 (electronic supplementary material, table S3), with an uncorrelated relaxed lognormal clock model for each partition and a Yule speciation tree prior. Analyses were run for 100 000 000 generations sampling every 10 000 generations, log files examined in Tracer 1.6 [47] for convergence (ESS > 200) and a maximum clade credibility (MCC) tree generated in TreeAnnotator 1.10.4 [46] with a conservative burn-in of 25%. We used three fossil calibrations with exponential distributions and an arbitrary mean of 5, and one geological calibration, all used by Agarwal *et al.* [48]: crown Gekkota (amber fossils from Myanmar; offset 99); most recent common ancestor (mrca) New Zealand Diplodactylidae (New Zealand diplodactylid material; offset 19); stem calibration for mrca *Pygopus* Merrem (*Pygopus hortulanus*; offset 23); divergence of *Phelsuma inexpectata* Mertens on Reunion from its closest relative on Mauritius *P. ornata* Gray (uniform prior, 0.05–5). Divergence times are presented in millions of years ago (Ma) and as median (95% highest posterior density) Ma.

We designated five discrete ancestral areas for reconstructions on the final timetree for the *H. mabouia* complex, which besides the Americas (New World) correspond to large-scale biogeographic divisions of Africa defined using a large dataset of plant and vertebrate species [49] and previously employed in the study of other African lizards [50]: Congolian, Madagascar (and associated islands), southern African and Zambezian regions (figure 1). Ancestral areas were estimated in RASP 4.2 [51] with a maximum of four ancestral areas using the Bayesian binary Markov chain Monte Carlo (MCMC) model (BBM). BBM analyses included fixed state frequencies, equal among site variation, temperature of 0.1 and 10 chains with a sample frequency of 1000 across 1 000 000 generations with a 10% burn-in; while dispersal–extinction–cladogenesis (DEC) analyses had no dispersal or range constraints. We interpret unambiguous reconstructions where one inferred ancestral area had at least twice as much support as the next most likely area.

## 2.4. Species distribution modelling

Species distribution models (SDMs) were built in MaxEnt 3.4.1 [52] using BIO1, BIO5, BIO6 and BIO12–BIO14 (average temperature and rainfall, and extremes of these, the same layers used by an earlier

predictive modelling study of invasive *Hemidactylus*—[53]) downloaded from WorldClim2 (http://www.worldclim.com/version2) at a 10 min resolution. Models used 'Auto features', Cloglog output format and a random test percentage of 10% with other settings at their defaults and were built using three different subsets of training localities: (i) the entire *H. mabouia* complex from Africa, (ii) *H. mabouia mabouia ss* from its putative native range in Africa (excluding Angolan and southern African localities as the species was historically absent from these regions; [34,36]), and (iii) *H. mabouia mabouia ss* from across Africa and the New World. Our goal was to contrast the SDMs built using our phylogenetically informed taxon subsets with each other to explore the questions: (i) does the realized spatial niche of *H. mabouia ss* in Africa allow it to inhabit the New World, (ii) is its absence from Zambezia a sampling artefact or linked to suitability, and (iii) what impact can incorrect taxonomy have on modelling the distribution of an invasive species? We evaluated the models using the area under the curve (AUC) for the test and training data on the receiver operating characteristic curve (AUC > 0.9).

# 3. Results

## 3.1. Phylogenetic relationships and species diversity

Individual gene trees (not shown) and the concatenated nuclear alignment all recover strong support (bootstrap support 93–100) for a monophyletic *Hemidactylus*, *Hemidactylus* + *Dravidogecko*, and all major global *Hemidactylus* clades (*sensu* [8,19]); though basal relationships within *Hemidactylus* remain poorly resolved (electronic supplementary material, figures S1 and S2). The ND2 and nuclear datasets also strongly support the monophyly of the *H. mabouia* complex within a well-supported 'African-Atlantic + H. mabouia' clade (figure 2; [8,36]). The only major discordance between the nuclear and ND2 data is in the placement of a *H. platycephalus* group specimen from Mozambique, sister to the *H. mabouia* complex in the nuclear tree and sister to the African-Atlantic clade in the ND2 tree. Additionally, relationships within the *H. mabouia* complex are not well supported in the nuclear trees and the description of relationships within the complex are based on the ND2 data.

Species delimitation analyses using the ND2 tree and a 5% ND2 divergence threshold converged on 20 putative species-level lineages within the *H. mabouia* complex (figure 2; electronic supplementary material, tables S5 and S6), with uncorrected pairwise ND2 divergence between putative species ranging from 5.5 to 26.1% (electronic supplementary material, table S6). The nuclear data were not informative and recovered a different subset of species and individuals (electronic supplementary material, figure S2 and table S5). Nominotypical *H. mabouia* or *H. mabouia ss* is a single species-level lineage that includes all New World and Congolian populations, as well as some peripheral populations in southern Africa and Zambezia, while the other 19 putative species (lineages) are more narrowly distributed (figure 1). Within the *H. mabouia* complex, a coastal Kenyan lineage (KEN) is deeply divergent from all other taxa, followed by a clade including three lineages restricted to south and southeastern Zimbabwe (ZIM4, ZIM5, ZIM6). The greatest genetic diversity is in a clade including eight putative species that are distributed in Zimbabwe, Mozambique and northeastern South Africa (MOZ1, MOZ2, SA1, SA2, ZAM, ZIM1, ZIM2, ZIM3), some occurring in near sympatry. This diverse clade as a whole is sister to a subclade including three lineages allied to *H. mercatorius* from Madagascar that are collectively sister to *H. mabouia ss*, and the sister group to these, comprising three lineages from Mozambique (*mabouia* 2, *mabouia* 3, *mabouia* 5) and one lineage (*mabouia* 4) with a disjunct distribution in Burundi and South Africa (presumably a translocated individual).

## 3.2. *Hemidactylus mabouia sensu stricto* phylogeography

A divergence of 3.0% in ND2 sequence data within *H. mabouia ss* separates two shallow subclades (A and B; figure 2; mean divergence within each 0.4–0.9%). The highest diversity within *H. mabouia ss* is in East Central Africa—basal splits separating unique haplotypes from Burundi and DRC in subclade A and Burundi and Uganda in subclade B (figure 2, electronic supplementary material, figure S1). Both subclades include one very widely distributed or invasive haplotype in Africa and the New World (A1, B1; electronic supplementary material, table S1). Subclade A has 21 haplotypes which are all Congolian singletons, except two from Southern Africa, one from the New World (Brazil), one from Zambezia (DRC) that is shared with a Congolian locality (Burundi), and finally, the invasive haplotype A1 is represented by 54 samples from 33 localities across the Congolian, New World,

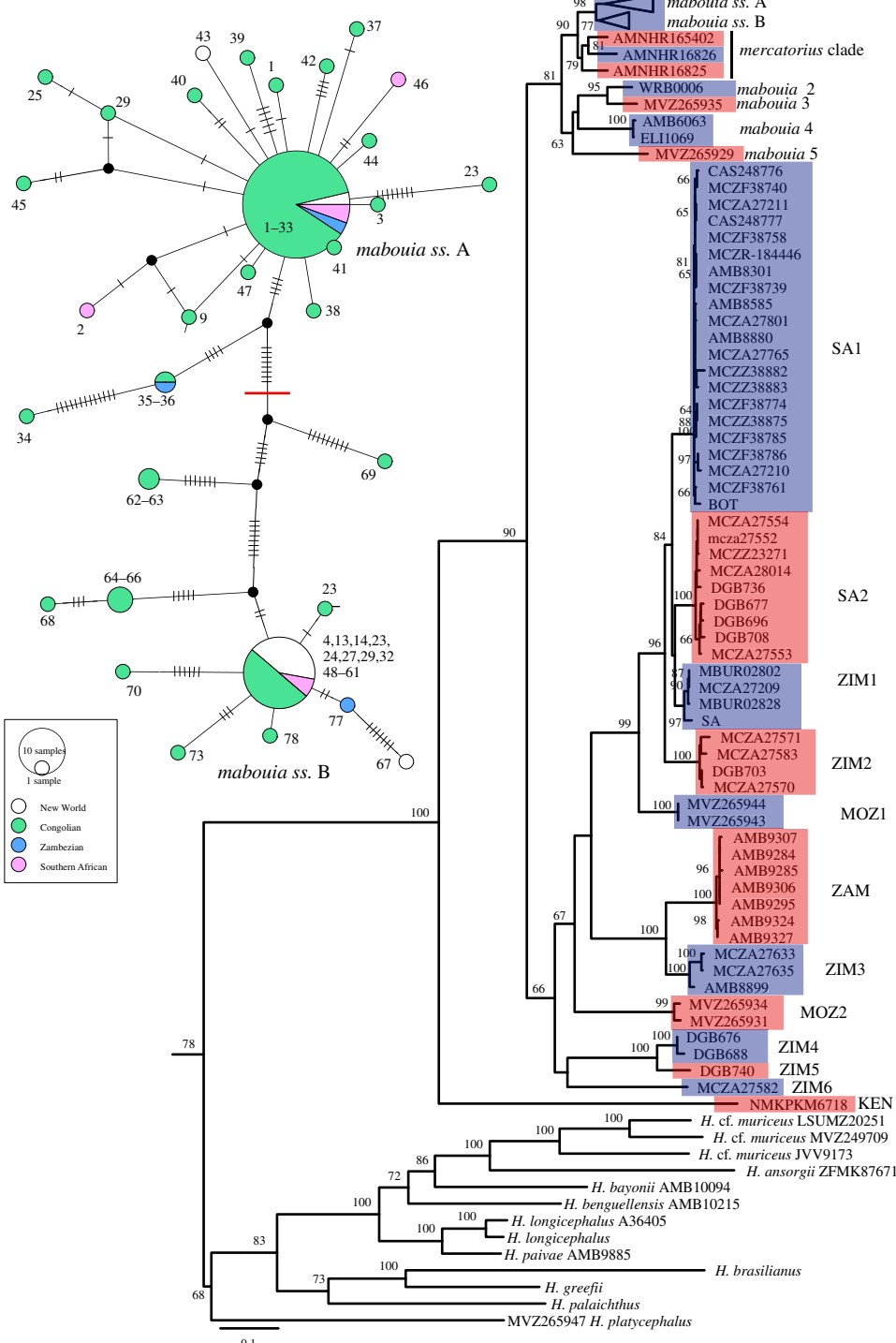

**Figure 2.** Main figure: maximum likelihood phylogeny of *Hemidactylus* with ND2 sequence data. Putative species from delimitation analyses are shaded with alternating pink and blue, bootstrap support/posterior probability is shown at nodes; *Hemidactylus* outside the African-Atlantic group + *H. mabouia* and outgroup taxa not shown (see electronic supplementary material, figure S1 for complete ND2 tree). Inset upper left: TCS haplotype network for *H. mabouia ss* using 892 nt of ND2 data. Nodes are scaled by haplotype frequency, numbers at nodes reference localities in electronic supplementary material, table S1, haplotypes are coloured based on the biogeographic realm they occupy (figure 1), hatch marks on connecting lines indicate mutations and black circles indicate unsampled intermediate haplotype states. The red line indicates the separation between the two main subclades of *H. mabouia ss*.

Southern African and Zambezian realms. Subclade B has 11 haplotypes that are all Congolian singletons except one from Zambezia (Mozambique), one from the New World (Trinidad and Tobago), two Congolian haplotypes represented by two and three samples each (two from Uganda; Burundi, DRC

and Uganda) and the final haplotype B1 which is represented by 24 samples from 21 localities in the Congolian, New World and Southern African realms.

The two invasive haplotypes range from western DRC and southern Angola to at least Ghana in Africa, and are sympatric at a number of localities. Of the 14 New World *H. mabouia ss* we sampled, two samples represent haplotype A1 and 10 haplotype B1, with a unique haplotype within each subclade, and both A1 and B1 are present in close proximity on Puerto Rico. In the Gulf of Guinea islands A1 is the most prevalent haplotype on São Tomé (all samples), Principe (all but one) and Bioko (all samples).

## 3.3. Divergence dating, ancestral area reconstructions and SDMs

Our estimates for the split between *Dravidogecko* and *Hemidactylus* 71 (82–61) Ma overlap with previously published estimates (electronic supplementary material, figure S3; [54]). The mrca of the African-Atlantic clade + the *H. mabouia* complex + *H. platycephalus* was estimated at 41 (48–35) Ma, and the mrca of the African-Atlantic clade at 33 (39–27) Ma. The mrca of the *H. mabouia* complex was dated at 21 (25–17) Ma, the Zimbabwean (ZIM4–ZIM6) and Zambezian-South African (MOZ1, MOZ2, SA1, SA2, ZAM, ZIM1, ZIM2, ZIM3) subclades each began diversifying 11 (13–9) Ma, within which the smaller South African subclade (SA1 + SA2) diverged from their sister group 3 (4–2) Ma and the mrca of the final Zambezian clade (*mabouia*2–*mabouia*5) is dated to 6 (7–5) Ma (figure 1). *H. mabouia ss* and *H. mercatorius* shared an mrca 3 (4–2) Ma and the split within the two clades of *H. mabouia* was within the last 1 (2–1) Ma.

The origin of the *H. mabouia* complex was reconstructed to have been in Zambezia, and apart from the smaller South African subclade (SA1 + SA2) that has a South African origin and the *H. mercatorius* group (three lineages) with a Malagasy origin, the ancestral areas at all other nodes were unequivocally reconstructed as Zambezian, except the mrca of *H. mabouia ss* which was reconstructed as widely distributed (figure 1). Other methods of ancestral area reconstruction (not shown) gave similar results except regarding the ancestor of *H. mabouia ss* + the *H. mercatorius* group which was reconstructed as being widely distributed.

The SDMs built using the three different subsets of locality data all had high AUC (0.96–0.98) and high predicted suitability across the known African and New World range as well as scattered regions in South and Southeast Asia (figure 3). The SDMs for *H. mabouia ss* from its putative native range in Africa and from all localities in Africa and the New World both had qualitatively similar results, with high suitability in coastal and tropical regions of Africa and the Americas and low suitability across Zambezia and southern Africa, though the latter model yielded higher predictions on tropical island habitats (figure 3). The SDM for the entire *H. mabouia* complex from Africa considerably overpredicted the distribution of *H. mabouia ss* in Africa, Madagascar, the Americas, and warm temperate and tropical regions across the globe, similar to the results of Rödder *et al.* [53] and Weterings and Vetter [55].

# 4. Discussion

## 4.1. Phylogeny and species diversity

Our unprecedented sampling of '*H. mabouia*' from across Africa and the New World reveal it is a species complex that probably originated in the Miocene in Zambezia. The complex may include as many as 20 species-level lineages, 14 of which are restricted to Zambezia, aside from the *H. mercatorius* complex in Madagascar (three putative species), two southern African lineages (near the Zambezian realm border) and the cosmopolitan *H. mabouia ss*; the latter being one of the most widely distributed terrestrial reptiles, established across sub-Saharan Africa and the Neotropics [6,19,29]. *Hemidactylus mabouia ss* shared a Zambezian mrca with the *H. mercatorius* complex and has had at least four independent colonizations of the New World, with two haplotypes identical to potential source populations in Africa and two unique New World haplotypes (similar to the results of [19]). Diversity within the *H. mabouia* complex is highly likely to remain underestimated as there are vast tracts of suitable habitat which remain unsampled. This has been exacerbated by the fact that many researchers have viewed '*H. mabouia*' as an uninteresting, largely commensal form, meaning that it may be underrepresented in collections from certain critical areas of southeastern Africa. Additionally, members of the *H. mabouia* complex are cryptic, with conserved morphology across most lineages within the complex ([37]; LMP Ceríaco, I Agarwal and AM Bauer 2021, unpublished data). Evaluating the true diversity and

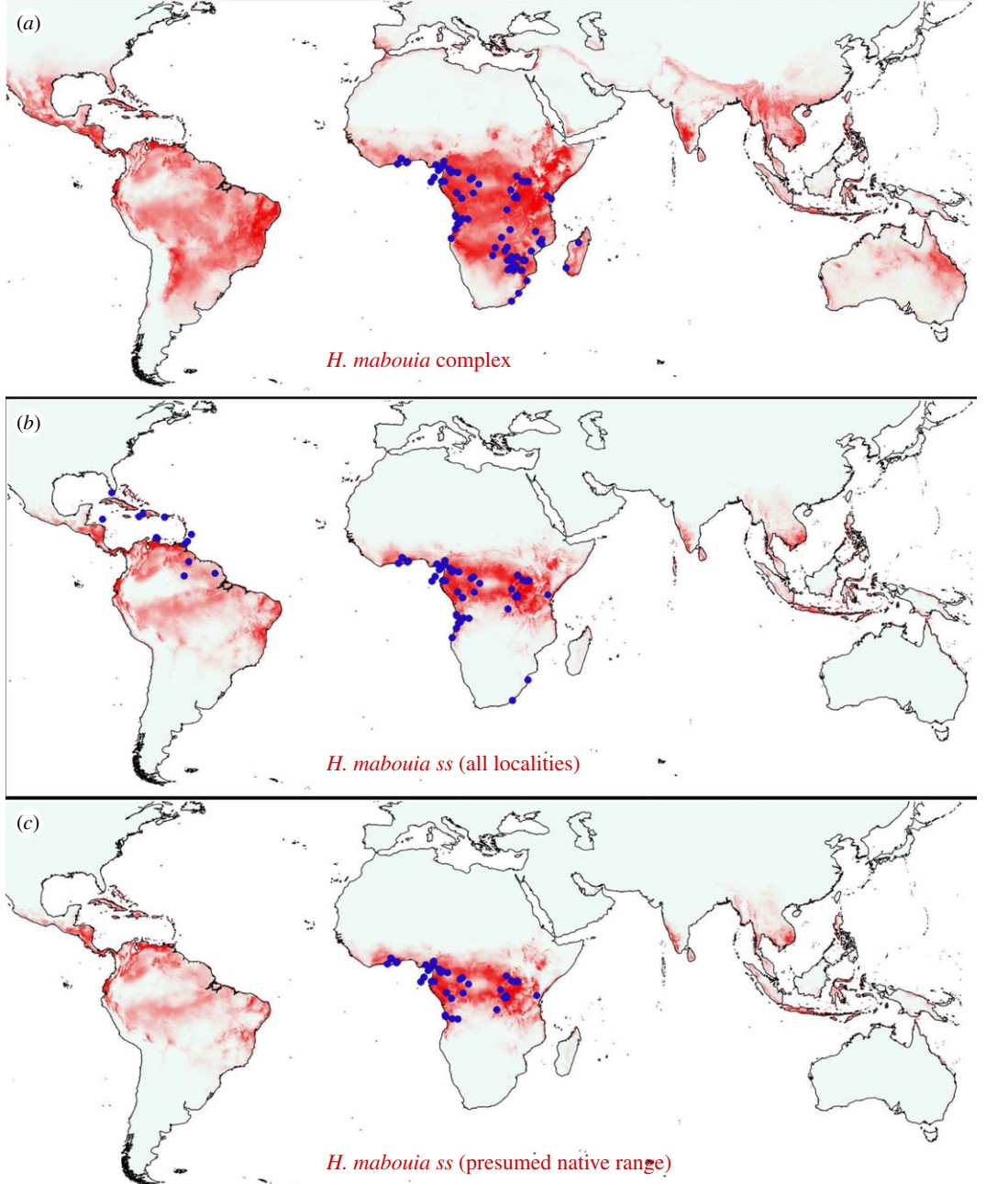

**Figure 3.** Maxent species distribution models using the Cloglog output for, (a) the H. mabouia complex, (b) all localities of H. mabouia ss and (c) only putative native localities of H. mabouia ss. Maps are scaled from low suitability (grey) to high suitability (red).

distribution of the group requires dedicated survey effort across Zambezia and along the border with the Congolian and Southern African realms in an integrative taxonomic framework.

## 4.2. Biogeography

*Hemidactylus mabouia ss* is primarily a tropical species, the bulk of distributional records from the Congolian realm, while the highest diversity within the *H. mabouia* complex is in southeastern Zambezia, and the *H. mercatorius* group is restricted to Madagascar; reflecting the broad biogeographic realms in Africa, which were erected using very different taxonomic groups [48]. We recovered a number of geographically restricted clades that began diversifying from the late Miocene to Pliocene, each of which includes multiple low–mid elevation lineages (distributional localities less than 800 m) and at least one high elevation lineage (greater than 1200 m); including a clade with three lineages in southeastern Zimbabwe,

a clade with three lineages in Mozambique within less than 150 km of each other, a clade with two lineages that extends from the Caprivi Strip in Namibia and Zimbabwe into Zambia, and a more broadly distributed clade with six lineages that extends from northern Botswana and northeastern South Africa to northern Mozambique, including two sister species in the northern Drakensberg of South Africa.

Other rupicolous lizards with high diversity in this topographically complex landscape include the geckos *Afroedura* [56] and *Lygodactylus* [57] and the cordylid *Platysaurus* [58]. Diversification within *Lygodactylus* and *Platysaurus* in this landscape also dates back to the Miocene–Pliocene [57,58], a time of global aridification and changing climate in Africa, with forest in the early Miocene replaced by C4 grasslands and a forest mosaic by the middle–late Miocene [59,60]. The inherently patchy distribution of rocky habitats and their isolation with aridification, besides diversification in montane regions has been implicated in the diversification of these groups [56–58]. Unfortunately, we lack even basic ecological information for members of the *H. mabouia* complex, at least some of which are potentially commensal (though apparently not invasive), and the presence of multiple species with overlapping distributions further confounds interpretations. The role of elevation does seem to be borne out, with a number of high elevation species nested within low elevation clades and potential montane diversification in the Mozambique and South African subclades. Much more sampling is needed across this vast, heterogeneous landscape to assess the diversity and distribution of members of the *H. mabouia* complex in order to understand its biogeographic history.

## 4.3. Taxonomic implications

The taxonomic and nomenclatural history of *H. mabouia* is convoluted, but has remained relatively stable since Loveridge's [61] revision of African *Hemidactylus*. The species was described based on West Indian populations by Moreau De Jonnès [25] in the early nineteenth century (type locality restricted by Stejneger [62] to 'St Vincent, Lesser Antilles'). One of the main problems regarding the taxonomic identity of *H. mabouia* is rooted in its original description. As noted by Kluge [5], Moreau De Jonnès' holotype, still extant in the collections of the Muséum national d'Histoire naturelle in Paris (accession number: MNHNP 6573), is not conspecific with what subsequent authors considered as *H. mabouia*, nor with our current interpretation of the species, but is rather allied to *H. angulatus*. In order to maintain taxonomic and nomenclatural stability, Kluge [5] recommended the maintenance of prevailing usage. This usage has been maintained for more than half a century but remains an impermanent solution and we plan a petition to the International Commission on Zoological Nomenclature to set aside the existing type and designate a neotype representing the species now universally associated with the nomen *Hemidactylus mabouia* (LMP Ceríaco, I Agarwal and AM Bauer, in preparation).

Interestingly, few names have been coined for such a widespread species complex, and most originate from specimens collected outside the natural distribution of the group. This can be explained by the fact that *H. mabouia sensu lato* has long been considered as a pan-African species (e.g. [63]), with a conservative morphology. Loveridge [61], Vanzolini [64], Kluge [5] and most recently Powell *et al.* [65] provided lists of synonymies for *H. mabouia*. Several of those included by Loveridge [61] have since been recognized as valid: *Hemidactylus mercatorius* Gray, 1842, from Madagascar; *Hemidactylus platycephalus* Peters, 1854, from Mozambique; *Hemidactylus tasmani* Hewitt, 1932, from Zimbabwe. Of the synonyms still listed by the latter authors, *Hemidactylus benguellensis* Bocage, 1893, from Angola has been recognized as a distinct species outside of the *H. mabouia* complex [36] and of the remaining four, *Gecko aculeatus* Spix, 1825, *Gecko armatus* Wied, 1825, *Gecko cruciger* Spix, 1825, and *Gecko incanescens* Wied, 1825, are from Brazil, and *Hemidactylus frenatus* var. *calabaricus* Boettger, 1879, is from Nigeria, all coming from areas that would suggest they should truly be synonyms of *H. mabouia ss*. Thus, aside from *H. mabouia* itself, only *H. mercatorius* and *H. tasmani* are applicable to other valid species in the *H. mabouia* complex, leaving most of the recently discovered lineages without available names. Besides the nomenclatural actions needed to stabilize and allocate the nomen *mabouia* to the Western African/invasive clades, taxonomic work is also needed to correctly allocate *tasmani* and *mercatorius* to one of the respective Zimbabwean and Madagascan genetic lineages. A critical taxonomic revision is, therefore, needed to clarify the species boundaries within the complex, provide new diagnoses for the already described species and describe all the others (LMP Ceríaco, I Agarwal and AM Bauer, in preparation).

## 4.4. How *Hemidactylus mabouia* conquered the world

### 4.4.1. Origin of *Hemidactylus mabouia ss* in Africa and the New World

Biogeographic analyses suggest a widespread ancestral area for *H. mabouia ss*, though the high haplotype diversity at the juncture of the Congolian and Zambezian realms and the fact that the two invasive

haplotypes make up most other distributional records is indicative of relatively recent expansion into western and southern Africa and the New World (figures 1 and 2; [32–34]). The earliest reported records of *H. mabouia ss* in the New World date from approximately 1637 to 1654, about 125 years after the commencement of the Atlantic slave trade. The earliest West Indian record of what is probably *H. mabouia* [24] is from St Kitts and was recorded a decade after the rise of sugar cultivation on the island (in 1643; [66]). A watercolour illustration made by Georg Marcgrave in Dutch-controlled northeastern Brazil between 1637 and 1644 is the first confirmation of the species' occurrence in mainland South America [67]. Beyond the middle of the seventeenth century, historical records relevant to the species distributions in both Africa and the New World are lacking.

Kluge [5] considered the likely origin of *H. mabouia* in the New World, and on the basis of perceived south to north clinal patterns in scalation features and its apparent absence from some islands on major trade routes, he speculated that it may have been derived from 'natural' overseas dispersal. Likewise, he postulated a similar origin for the American *H. brookii* (= *H. angulatus*) group species *H. haitianus*, *H. leightoni* and *H. palaichthus* [5] as did Vanzolini [64,68,69]. The latter conclusion has been partly borne out by molecular phylogenetic studies that recognize an African-Atlantic clade of *Hemidactylus* with intercontinental species divergences into the Palaeogene [8,9,19]. The origin of *H. mabouia* in the New World, however, remains inconclusive. Our BEAST analysis cannot distinguish between human-mediated transport and very recent 'natural' overseas dispersal, although the shared occurrence of two identical invasive haplotypes in the New World and West and Central African populations (figures 1–2; electronic supplementary material, table S1) are consistent with recent expansion into these parts of Africa and multiple colonizations of the Americas. Whether the widespread occurrence of these haplotypes (especially B1) in the New World represents movement within the Western Hemisphere or multiple instances of trans-Atlantic colonization by the same haplotype can, likewise, not be determined. If Kluge's [5] claim of a morphological cline in the New World is correct, it would be too recent for significant substructuring to have evolved and any phenotypic variation is likely to be along an environmental gradient [70]. It is most likely to reflect one based on plasticity rather than heritability.

We suggest, as did Vanzolini [64], that early globalization is a plausible cause of the initial spread of *H. mabouia* across the Atlantic. Transport from Africa to the New World was part of the so-called triangular trade and was dominated by the transport of enslaved people bound for colonial ports from Uruguay to British North America. Of 31 821 documented slave voyages from 1525 to 1875 (www.slavevoyages.org), nearly 2000 had already occurred by 1650 (the approximate first documentation of *H. mabouia* in the New World), the majority originating in West Central Africa [71]. The vast majority of enslaved people during this period were disembarked in Spanish colonies in the Americas, chiefly the mainland areas surrounding the Caribbean Sea as well as the Greater and Lesser Antilles, virtually all of which were Spanish territory prior to 1625. This was followed, at a distant second, by the Pernambuco region of northeastern Brazil (www.slavevoyages.org), where Marcgrave illustrated *H. mabouia* in the mid-seventeenth century [67].

Patterns of haplotype distribution in mainland Africa also suggest a very recent spread of the most common haplotypes, with no geographical signal outside of the East African Rift Valley, where basal members of the *H. mabouia ss* clade are found. The widespread occurrence of the A1 and B1 haplotypes throughout West and West Central Africa, and the preponderance of known localities in and around disturbed area (see below) is likewise consistent with human-mediated transport. This could have taken place at any time in the past, but as early as the Upper Palaeolithic there were human migrations into both West Central and West Africa that could have provided the means for *H. mabouia* to spread [72]. However, trade connections throughout the continent would have provided opportunities for the movement of this and other invasive species up to the present.

### 4.4.2. Opportunity, ecological tolerance or a bit of both?

*Hemidactylus* are known to outcompete other geckos in topographically simple habitats with clumped resources (such as below night lights in urban settings; [73]), and *H. mabouia ss* is specifically known to outcompete and directly predate on both native and other invasive geckos (e.g. [26,31,74,75]). *Hemidactylus mabouia ss* is a scansorial generalist with a preference for open, human-dominated habitats [76], though it is also found in natural habitats in both the New World and Africa [30]. Of 70 of our unique *H. mabouia ss* localities in Africa, only 14 were from either forest or field stations within forest.

Numerous species of narrowly endemic scansorial *Hemidactylus*, including other members of the *H. mabouia* complex also use edificarian habitats, though not necessarily feeding close to lights (e.g. [54,77,78]), but do not appear to have colonized areas far outside their distributional range or to have

become invasive. A possible exception is *mabouia* 4, which we have identified only from one locality each in Burundi and South Africa which are approximately 3000 km apart and less than 0.5% divergent in ND2 sequence data (unpublished 16 s sequences of 415 nt for these two samples are identical to each other and two published sequences from Durban and Port Elizabeth, South Africa; LMP Ceríaco, I Agarwal and AM Bauer 2021, unpublished data). In this instance, long-distance colonization has taken place, but there is no evidence of invasiveness or even confirmation of local establishment—though our sampling is limited. Such cases are common among commensal geckos, particularly in the age of container shipments, which can transport geckos or their eggs between almost any two places on Earth in 30–45 days or less. The sporadic occurrence of such long-distance translocations says something about commensalism but nothing about invasiveness. Our sampling suggests that just 2/20 putative species in the *H. mabouia* complex, *H. mabouia* ss and *H. mabouia* 4 have been moved around by people, and only a single species is invasive; and even within each of the two subclades of *H. mabouia* ss, just 1/21 and 1/11 haplotypes seem to be invasive.

SDMs built for *H. mabouia* ss from its putative native range show that this is largely a tropical species, with high predicted suitability in the Neotropics indicating the realized niche of *H. mabouia* ss in Africa allowing it to inhabit the New World; making it a strong candidate for establishment [79]. The models also predict low suitability in Zambezia and southern Africa—regions from which the species is largely absent—suggesting a real biological or physiological limit rather than a sampling artefact. The model for the entire *H. mabouia* complex overpredicts, including areas outside the range of occurrence of *H. mabouia* ss, and highlights the importance of taxonomy and understanding the natural variation of a species in trying to predict its spread (e.g. [80–83]). Geographical range expansion in the case of invasive, commensal geckos has been shown to be linked to the expansion of the realized niche [84], which does not seem to be the case for *H. mabouia*. This is indicative that biotic interactions may be the limiting factor in the distribution of the species, or may reflect the lack of opportunity for dispersal to novel climates. Much of the diversity of the *H. mabouia* complex is in the Zambezian and southern African realms, where *H. mabouia* ss is largely absent. In fact, though the two subclades of *H. mabouia* ss are sympatric at a number of localities in Africa and the New World, *H. mabouia* ss was not sympatric with another member of the *H. mabouia* complex in our sampling. It is unclear what specific aspects of the ecology of the invasive clades of *H. mabouia* ss have allowed them to establish widely. Three potential explanations that need not be mutually exclusive linked to thermoregulation are (i) that urban areas are substantially warmer and potentially provide warmer microhabitats than surrounding natural habitats [85], (ii) that *H. mabouia* ss has been able to shift its thermal biology to invade cooler climes (as in the tropical invader *H. frenatus* in temperate Australia; [86]), and (iii) that realized climatic niches of invasive herpetofaunal species tend to be conserved along maximum temperatures, with more lability in other aspects of climate [87]. Invasive geckos are known to displace native species based on (larger) body size, faster locomotion and other traits linked to aggressive or dominant behaviour (e.g. [2,74,88,89]). Understanding the factors that allow *H. mabouia* ss to have become invasive would involve contrasting it and other members of the *H. mabouia* clade in behaviour, ecology and thermal physiology from across the current range of *H. mabouia* ss.

# 5. Conclusion

*Hemidactylus mabouia* is a deeply divergent species complex that originated in the Miocene in Zambezia with at least 20 species-level lineages that include multiple commensal species. Just one species-level lineage, *H. mabouia* ss, appears to have evolved the traits necessary to become invasive, and extensive human movements within Africa and across the Atlantic to suitable habitats provided the opportunity for this species to conquer the world. This shows how rare 'invasiveness' is as a biological capacity and also that *H. mabouia* has been a successful colonist within the African continent, not just on islands in the relatively gecko-depauperate New World. There are important implications for the management of invasive species—an accurate assessment of diversity within taxa being of critical importance. *Hemidactylus* is a useful model system to understand the evolutionary origins of invasiveness, including many invasive species sister to non-invasive species with restricted distributions.

Ethics. This work has been approved by and complies with the requirements of all relevant institutional and government authorities. Animal tissues obtained directly for this study were collected under animal ethics protocols numbers AS 14–70 and 1866 from Villanova University. Specimens and tissues were imported into the United States under collecting and export permits from Angola, Namibia, South Africa and Zimbabwe and were cleared through the United States Fish and Wildlife Service. Additional material was obtained through museum loans.

Data accessibility. DNA sequences are available from Genbank using the accession numbers listed in the electronic supplementary material, tables S1 and S4. The mtDNA alignment, concatenated nuclear alignment and divergence dating alignment and trees are available at TreeBASE (accession number 28137; http://purl.org/phylo/treebase/phylows/study/TB2:S28137?x-access-code=b26c9b22c83db255344e37b6e38729d1&format=html). Climate data and MaxEnt input files are available at http://dx.doi.org/10.5281/zenodo.4596131.

The data are provided in electronic supplementary material [90].

Authors' contributions. A.M.B. and M.M. conceived and planned the study; A.M.B., I.A., L.M.P.C. and M.M. conducted fieldwork; I.A., M.M. and T.R.J. generated molecular sequence data; I.A. performed analyses and wrote the manuscript with critical inputs and revisions from A.M.B. and L.M.P.C.; all authors contributed to the revision of the article and approved the final version of this manuscript.

Competing interests. We declare that we have no competing interests.

Funding. This work was supported by the US National Science Foundation (grants EF 1241885, subaward 13-0632 and DEB 1556255) to A.M.B.

Acknowledgements. We would like to thank Álvaro (Varito) Baptista and his team for all the assistance and outstanding support during the field surveys, and for assistance in the field, Suzana Bandeira, Mariana Marques, Jens Vindum, Edward Stanley and David Blackburn. This work was carried out in collaboration with the Instituto Nacional da Biodiversidade e Áreas de Conservação (INBAC) from the Ministry of Environment of Angola which provided institutional and logistical support, as well as the necessary permits for carrying out this research. We thank the following museums and curators for access to tissue samples in their care: Lauren Scheinberg from the California Academy of Sciences (CAS); David Kirzirian and Lauren Vonnhame from American Museum of Natural History (AMNH); Steve Rogers from the Carnegie Museum of Natural History (CM); Ned Gilmore from the Academy of Natural Sciences of Drexel University (ANSP); Alan Resetar from the Field Museum of Natural History (FMNH); Rayna Bell and Addison Wynn from the National Museum of Natural History, Smithsonian Institution (USNM); José Rosado from the Museum of Comparative Zoology (MCZ); Carol Spencer and Jim McGuire from the Museum of Vertebrate Zoology (MVZ); Adam Leaché from the University of Washington Burke Museum (UWBM); Eli Greenbaum from University of Texas at El Paso (UTEP); Larry Lee Grismer from La Sierra University Herpetological Collection (LSUHC); Bryan Stuart from North Carolina State Museum (NCSM); Donna Dittman from Louisiana State University Museum of Zoology (LSUMZ); Patrick Campbell from the Natural History Museum (BMNH); Annemarie Ohler and Marc Cugnet from the Muséum national d'Histoire naturelle (MNHN-RA); Andreas Schmitz from the Muséum d'Histoire naturelle (MHNG); Arnaud Maeder and Nicolas Margraf from the Musée d'Histoire naturelle de La Chaux-de-Fonds (MHNC); the Michael Franzen from Zoologische Staatssammlung München (ZSM); Jakob Hallermann from, Zoologisches Museum Hamburg (ZMH); Rainer Gunther and Frank Tillack from the Museum für Naturkunde (ZMB); Dennis Rödder and Wolfgang Böhme from Zoologische Forschungsmuseum Alexander Koenig (ZFMK); Silke Schweiger and Georg Gassner from the Naturhistorisches Museum (NHMW); Hussam Zaher from Museu de Zoologia da Universidade de São Paulo (MZUSP); Belmira Gumbe and Ana Lavres from the Museu de Nacional de História Natural (MNHNL); Ilunga André from the Museu Regional do Dundo (MD); Lauretta Mahlangu from the Ditsong National Museum of Natural History (TM), Pretoria, South Africa, and William R. Branch from Port Elizabeth Museum (PEM). Miguel Trefaut Rodrigues provided valuable insights regarding the historical presence of *H. mabouia* in Brazil.

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
