## [Peer Review File · Royal Society Open Science]

Review History

RSOS-210749.R0 (Original submission)

Review form: Reviewer 1

Is the manuscript scientifically sound in its present form?

Yes

Are the interpretations and conclusions justified by the results?

Yes

Is the language acceptable?

Yes

Do you have any ethical concerns with this paper?

No

Have you any concerns about statistical analyses in this paper?

No

Recommendation?

Accept with minor revision (please list in comments)

Comments to the Author(s)

I enjoyed this paper. It was fascinating to see the previously unrecognized genetic diversity in the *H. mabouia* complex, particularly in light of their worldwide colonization. It was also interesting to see the relevance of those findings in the results of the niche modeling. Interesting work!

There are two main things I'd like to be addressed and or considered prior to publication, and I also have a few comments on specific lines.

1. More time should be invested in making supplementary figures valuable and interpretable. Both figures S2 and S3 would be improved. S3 really needs complete taxonomic names for all the included species. Figure S2 could use some sort of labeling approach that allows the reader to ability to interpret the results in light of the mtDNA data (and figures 1 and 2).
2. I'd suggest more consideration of how you refer to as "putative species". It's well understood now that deep mtDNA structure does not always equal cryptic species, and that mtDNA clades can't simply be referred to as "putative species" in the context of a lack of really informative nuclear data. Even if you really prefer to interpreting them as such, some qualification is needed as to the potential pitfalls with that. I think you can address this issue with relatively minor edits, but it should be addressed. E.g., line 261 "at least 20 species-level lineages". Maybe. But the literature now how plenty of examples where mtDNA divergence of >5% does not mean different "species-level lineages".

Minor comments with suggested edits (or at least points of consideration):

line 208 low is relative; suggest just stating that the two subclades are 3% divergent

line 367 "Our BEAST analysis cannot distinguish between human-mediated transport and very recent 'natural' overseas dispersal" Are you being overly cautious here? I think the data are consistent with human mediated invasion. Also, that's how you treat them later in the paper.

lines 378, 382 use the term "enslaved people" rather than slaves

432 "and even within the two subclades of *H. mabouia* ss, just 1/21 and 1/11 haplotypes seem to be invasive" I'd recommend some serious caution about over interpreting mitochondrial the data here. Mitochondrial haplotypes don't necessarily equal distinct lineages, and to infer that they represent the only "lineages" that have colonized (or have the ability to) is overly simplistic.

Line 441 "It is unclear what specific aspects of the ecology of the invasive clades of *H. mabouia* ss have allowed them to establish in novel climates." Doesn't the niche modeling indicate that they are NOT established in novel climates? I.e., niche modeling of *H. mabouia* ss from their native range predicts where they're found in the western hemisphere because the climatic conditions in the two places are similar enough? Lines 441-451 could use some tightening up.

Review form: Reviewer 2

Is the manuscript scientifically sound in its present form?

Yes

Are the interpretations and conclusions justified by the results?

Yes

Is the language acceptable?

Yes

Do you have any ethical concerns with this paper?

No

Have you any concerns about statistical analyses in this paper?

No

Recommendation?

Accept with minor revision (please list in comments)

Comments to the Author(s)

This is an important and valuable contribution clarifying the phylogeography and providing an initial taxonomic resolution of a very widespread but very poorly studied group of commensal geckos. The study is based on a large amount of new DNA sequences from previously unstudied samples, thoroughly analysed and illustrated. The analyses are adequate and correctly interpreted. Overall, I have only few suggestions for manuscript revision – of course there is a plethora of additional analyses that one could envisage, like separate analyses of each nuclear marker for improved species delimitation (but for this more samples of each lineage would need to be sequenced), more explicit model testing in terms of statistical phylogeography, etc – but most of these make little sense with the data set at hand and would only serve to unnecessarily inflate the paper. Overall, this is a great study absolutely meriting publication, and I look forward to seeing it published in *RSOS*.

Some of my main concerns relate to the summary where some aspects are presented too simplistic while in the main text they are usually discussed with the adequate care.

First sentence of Summary, and first sentence of Introduction: Be gentle with invasive species (of geckos)! Yes it is true that invasive species can become a severe threat! This is particularly true for introduced mammals predatory islands, or highly competitive weeds on islands, and sometimes for toxic species such as the cane toad. However, the vast majority of invasive species do no harm and sometimes even can enrich ecosystems (see the quite enlightening book of Fred Pearce "The New Wild: Why Invasive Species Will Be Nature's Salvation" – although there are good reasons to disagree with many of his statements).

But most importantly here, are there examples of invasive geckos that would constitute a true problem anywhere? I think there are indeed some problems with introduced *Phelsumas* on the Mascarenes, and maybe there are some other examples which I expect will always be in fragile island ecosystems. This should be explained in a more differentiated way – like it is now, a naive reader could think, invasive geckos are one of the biggest threats to global biodiversity, while certainly, the biggest threat continues being habitat destruction!

line 21 and 24: please be consistent and here also use "putative species-level lineages". You use this almost everywhere else in the manuscript, but please double check. This is an important distinction – you are here not presenting any fully convincing, integrative evidence for any of these putative species, although of course I agree they probably all are distinct at the species level.

line 23 (also line 260): also here, allow for uncertainty! You cannot state that these lineages originated in the Miocene ... but of course you could write that your analyses suggest such an age, or that you hypothesize this based on your analysis, or similar.

421-422: Another quite strong statement! Yes, you have sequenced many specimens, but not enough for such a strong statement. Just add a "may" or change the phrasing otherwise!

399: section Opportunity, ecological tolerance or a bit of both?

I would suggest two additional factor here (maybe to be listed in lines 444-451 or elsewhere in the section): behaviour and genetic variation! I am convinced that for many cases of lizards, and I suspect for many other invasive animals, aggressiveness and "dominant" behaviour are extremely important for their success as competitors. This in turn might be related to (not only intraspecific, but also intraindividual) genetic variation, a factor that might well be determinant to the adaptational capacity of an invasive propagule. The beauty of these two factors is that they could be easily tested, by behavioural experiments comparing different *Hemidactylus* species, and by genomic studies. I am pretty convinced successful invaders will on average turn out to be rather aggressive and highly active species, of course also with short generation times, and with high genetic variation mediating adaptability.

Caption to Fig. 1: Please explain on what data the timetree is based. I suppose the concatenated nuclear + mitochondrial data?

Decision letter (RSOS-210749.R0)

Dear Dr Agarwal,

On behalf of the Editors, we are pleased to inform you that your Manuscript RSOS-210749 "How the African house gecko (*Hemidactylus mabouia*) conquered the world" has been accepted for publication in Royal Society Open Science subject to minor revision in accordance with the referees' reports. Please find the referees' comments along with any feedback from the Editors below my signature.

Please submit your revised manuscript and required files (see below) no later than 7 days from today's (ie 05-Jul-2021) date. Note: the ScholarOne system will 'lock' if submission of the revision is attempted 7 or more days after the deadline. If you do not think you will be able to meet this deadline please contact the editorial office immediately.

Kind regards,
Royal Society Open Science Editorial Office
Royal Society Open Science

on behalf of Dr Polly Campbell (Associate Editor) and Kevin Padian (Subject Editor)
openscience@royalsociety.org

Associate Editor Comments to Author (Dr Polly Campbell):

Both reviewers are very positive about the value of this work. I agree with them and also appreciated the clarity with which it is presented. The reviewers also provide useful and important suggestions for improvement. I particularly encourage the authors to be more cautious and consistent in their language when making inferences about what is and is not a distinct species based on mtDNA haplotype divergence.

Additional minor requested edits

L27: suggest, not suggests

L84: suggest "estimate" rather than "determine"

L148-151: First part of sentence refers to biogeographic divisions in Africa but the Americas are included in the list. Also, delete one "and" on L149.

L192: add "with" before uncorrected.

L201-202: suggest, "This diverse clade..." or something similar rather than "This species-rich clade..." I also suggest breaking this sentence in two or revising for clarity. It's currently hard to follow from start to finish.

L313: is, not are

L374-375: Please justify and provide a citation for the statement about the morphological cline: "it is most likely to reflect one based on plasticity rather than heritability."

Reviewer comments to Author:

Reviewer: 1

Comments to the Author(s)

I enjoyed this paper. It was fascinating to see the previously unrecognized genetic diversity in the *H. mabouia* complex, particularly in light of their worldwide colonization. It was also interesting to see the relevance of those findings in the results of the niche modeling. Interesting work!

There are two main things I'd like to be addressed and or considered prior to publication, and I also have a few comments on specific lines.

1. More time should be invested in making supplementary figures valuable and interpretable. Both figures S2 and S3 would be improved. S3 really needs complete taxonomic names for all the included species. Figure S2 could use some sort of labeling approach that allows the reader to ability to interpret the results in light of the mtDNA data (and figures 1 and 2).
2. I'd suggest more consideration of how you refer to as "putative species". It's well understood now that deep mtDNA structure does not always equal cryptic species, and that mtDNA clades can't simply be referred to as "putative species" in the context of a lack of really informative nuclear data. Even if you really prefer to interpreting them as such, some qualification is needed as to the potential pitfalls with that. I think you can address this issue with relatively minor edits, but it should be addressed. E.g., line 261 "at least 20 species-level lineages". Maybe. But the literature now how plenty of examples where mtDNA divergence of >5% does not mean different "species-level lineages".

Minor comments with suggested edits (or at least points of consideration):

line 208 low is relative; suggest just stating that the two subclades are 3% divergent

line 367 "Our BEAST analysis cannot distinguish between human-mediated transport and very recent 'natural' overseas dispersal" Are you being overly cautious here? I think the data are consistent with human mediated invasion. Also, that's how you treat them later in the paper.

lines 378, 382 use the term "enslaved people" rather than slaves
 432 "and even within the two subclades of *H. mabouia* ss, just 1/21 and 1/11 haplotypes seem to be invasive" I'd recommend some serious caution about over interpreting mitochondrial the data here. Mitochondrial haplotypes don't necessarily equal distinct lineages, and to infer that they represent the only "lineages" that have colonized (or have the ability to) is overly simplistic.
 Line 441 "It is unclear what specific aspects of the ecology of the invasive clades of *H. mabouia* ss have allowed them to establish in novel climates." Doesn't the niche modeling indicate that they are NOT established in novel climates? I.e., niche modeling of *H. mabouia* ss from their native range predicts where they're found in the western hemisphere because the climatic conditions in the two places are similar enough? Lines 441-451 could use some tightening up.

Reviewer: 2

Comments to the Author(s)

This is an important and valuable contribution clarifying the phylogeography and providing an initial taxonomic resolution of a very widespread but very poorly studied group of commensal geckos. The study is based on a large amount of new DNA sequences from previously unstudied samples, thoroughly analysed and illustrated. The analyses are adequate and correctly interpreted. Overall, I have only few suggestions for manuscript revision – of course there is a plethora of additional analyses that one could envisage, like separate analyses of each nuclear marker for improved species delimitation (but for this more samples of each lineage would need to be sequenced), more explicit model testing in terms of statistical phylogeography, etc – but most of these make little sense with the data set at hand and would only serve to unnecessarily inflate the paper. Overall, this is a great study absolutely meriting publication, and I look forward to seeing it published in *RSOS*.

Some of my main concerns relate to the summary where some aspects are presented too simplistic while in the main text they are usually discussed with the adequate care.

First sentence of Summary, and first sentence of Introduction: Be gentle with invasive species (of geckos)! Yes it is true that invasive species can become a severe threat! This is particularly true for introduced mammals predatory islands, or highly competitive weeds on islands, and sometimes for toxic species such as the cane toad. However, the vast majority of invasive species do no harm and sometimes even can enrich ecosystems (see the quite enlightening book of Fred Pearce "The New Wild: Why Invasive Species Will Be Nature's Salvation" – although there are good reasons to disagree with many of his statements).

But most importantly here, are there examples of invasive geckos that would constitute a true problem anywhere? I think there are indeed some problems with introduced *Phelsumas* on the Mascarenes, and maybe there are some other examples which I expect will always be in fragile island ecosystems. This should be explained in a more differentiated way – like it is now, a naive reader could think, invasive geckos are one of the biggest threats to global biodiversity, while certainly, the biggest threat continues being habitat destruction!

line 21 and 24: please be consistent and here also use "putative species-level lineages". You use this almost everywhere else in the manuscript, but please double check. This is an important distinction – you are here not presenting any fully convincing, integrative evidence for any of these putative species, although of course I agree they probably all are distinct at the species level.

line 23 (also line 260): also here, allow for uncertainty! You cannot state that these lineages originated in the Miocene ... but of course you could write that your analyses suggest such an age, or that you hypothesize this based on your analysis, or similar.

421-422: Another quite strong statement! Yes, you have sequenced many specimens, but not enough for such a strong statement. Just add a "may" or change the phrasing otherwise!

399: section Opportunity, ecological tolerance or a bit of both?

I would suggest two additional factor here (maybe to be listed in lines 444-451 or elsewhere in the section): behaviour and genetic variation! I am convinced that for many cases of lizards, and I suspect for many other invasive animals, aggressiveness and "dominant" behaviour are extremely important for their success as competitors. This in turn might be related to (not only intraspecific, but also intraindividual) genetic variation, a factor that might well be determinant to the adaptational capacity of an invasive propagule. The beauty of these two factors is that they could be easily tested, by behavioural experiments comparing different *Hemidactylus* species, and by genomic studies. I am pretty convinced successful invaders will on average turn out to be rather aggressive and highly active species, of course also with short generation times, and with high genetic variation mediating adaptability.

Caption to Fig. 1: Please explain on what data the timetree is based. I suppose the concatenated nuclear + mitochondrial data?

===PREPARING YOUR MANUSCRIPT===

===PREPARING YOUR REVISION IN SCHOLARONE===

To revise your manuscript, log into <https://mc.manuscriptcentral.com/rsos> and enter your Author Centre - this may be accessed by clicking on "Author" in the dark toolbar at the top of the

page (just below the journal name). You will find your manuscript listed under "Manuscripts with Decisions". Under "Actions", click on "Create a Revision".

<https://royalsociety.org/journals/authors/author-guidelines/#supplementary-material> to include a suitable title and informative caption. An example of appropriate titling and captioning may be found at https://figshare.com/articles/Table_S2_from_Is_there_a_trade-off_between_peak_performance_and_performance_breadth_across_temperatures_for_aerobic_sc_ope_in_teleost_fishes_/3843624.

Author's Response to Decision Letter for (RSOS-210749.R0)

See Appendix A.

Decision letter (RSOS-210749.R1)

Dear Dr Agarwal,

I am pleased to inform you that your manuscript entitled "How the African house gecko (*Hemidactylus mabouia*) conquered the world" is now accepted for publication in Royal Society Open Science.

on behalf of Dr Polly Campbell (Associate Editor) and Kevin Padian (Subject Editor)
openscience@royalsociety.org

Appendix A

Editorial comments:

Additional minor requested edits

L27: suggest, not suggests

L84: suggest "estimate" rather than "determine"

L192: add "with" before uncorrected.

L313: is, not are

Response: Incorporated all minor edits

L148-151: First part of sentence refers to biogeographic divisions in Africa but the Americas are included in the list. Also, delete one "and" on L149.

Response: edited for clarity.

L201-202: suggest, "This diverse clade..." or something similar rather than "This species-rich clade..."

I also suggest breaking this sentence in two or revising for clarity. It's currently hard to follow from start to finish.

Response: edited for clarity.

L374-375: Please justify and provide a citation for the statement about the morphological cline: "it is most likely to reflect one based on plasticity rather than heritability."

Response: This has been reworded and a citation added "If Kluge's (1969) claim of a morphological cline in the New World is correct, it would be too recent for significant substructuring to have evolved and any phenotypic variation is likely to be along an environmental gradient (Kaliontzopoulou et al. 2018)." Kaliontzopoulou, A., Pinho, C., and Martínez-Freiría, F. 2018. Where does diversity come from? Linking geographical patterns of morphological, genetic, and environmental variation in wall lizards. *BMC Ecology and Evolution* **18**, 124 (2018).

<https://doi.org/10.1186/s12862-018-1237-7>

Reviewer comments to Author:

Reviewer: 1

Comments to the Author(s)

I enjoyed this paper. It was fascinating to see the previously unrecognized genetic diversity in the *H. mabouia* complex, particularly in light of their worldwide colonization. It was also interesting to see the relevance of those findings in the results of the niche modeling. Interesting work!

There are two main things I'd like to be addressed and or considered prior to publication, and I also have a few comments on specific lines.

1. More time should be invested in making supplementary figures valuable and interpretable. Both figures S2 and S3 would be improved. S3 really needs complete taxonomic names for all the included species. Figure S2 could use some sort of labeling approach that allows the reader to ability to interpret the results in light of the mtDNA data (and figures 1 and 2).

Response: Complete taxonomic names have been added to Fig. S2. Complete taxonomic names are already in Fig. S3, apart from the composite *Lialis*. Unfortunately it is not possible to rename Fig S2 that allows the reader to interpret in the context of the mtDNA data as the same exact clades are not recovered.

2. I'd suggest more consideration of how you refer to as "putative species". It's well understood now that deep mtDNA structure does not always equal cryptic species, and that mtDNA clades can't simply be referred to as "putative species" in the context of a lack of really informative nuclear data. Even if you really prefer to interpreting them as such, some qualification is needed as to the potential pitfalls with that. I think you can address this issue with relatively minor edits, but it should be addressed. E.g., line 261 "at least 20 species-level lineages". Maybe. But the literature now how plenty of examples where mtDNA divergence of >5% does not mean different "species-level

lineages".

Response: We have cited the gecko examples where mtDNA divergence of ~5% is indicative of species level divergence (line 122) and have also made some edits as suggested (line 261, 24)

Minor comments with suggested edits (or at least points of consideration):

line 208 low is relative; suggest just stating that the two subclades are 3% divergent

Response: done

line 367 "Our BEAST analysis cannot distinguish between human-mediated transport and very recent 'natural' overseas dispersal" Are you being overly cautious here? I think the data are consistent with human mediated invasion. Also, that's how you treat them later in the paper.

Response: The rest of the sentence qualifies this statement, and it is based on this that we consider they were likely moved with people.

lines 378, 382 use the term "enslaved people" rather than slaves

Response: edited

432 "and even within the two subclades of *H. mabouia* ss, just 1/21 and 1/11 haplotypes seem to be invasive" I'd recommend some serious caution about over interpreting mitochondrial the data here. Mitochondrial haplotypes don't necessarily equal distinct lineages, and to infer that they represent the only "lineages" that have colonized (or have the ability to) is overly simplistic.

Response: While it is true mitochondrial haplotypes do not necessarily represent distinct lineages, our data merely suggest that only a few haplotypes of the large number within this species complex as well as single species appear invasive.

Line 441 "It is unclear what specific aspects of the ecology of the invasive clades of *H. mabouia* ss have allowed them to establish in novel climates." Doesn't the niche modeling indicate that they are NOT established in novel climates? I.e., niche modeling of *H. mabouia* ss from their native range predicts where they're found in the western hemisphere because the climatic conditions in the two places are similar enough? Lines 441-451 could use some tightening up.

Response: The niche modelling results do in fact suggest that they have not established in completely novel climates but that their realised niche allows them to inhabit those regions. We have rephrased this to what has allowed them to establish widely: "It is unclear what specific aspects of the ecology of the invasive clades of *H. mabouia* ss have allowed them to establish widely."

Reviewer: 2

Comments to the Author(s)

This is an important and valuable contribution clarifying the phylogeography and providing an initial taxonomic resolution of a very widespread but very poorly studied group of commensal geckos. The study is based on a large amount of new DNA sequences from previously unstudied samples, thoroughly analysed and illustrated. The analyses are adequate and correctly interpreted. Overall, I have only few suggestions for manuscript revision – of course there is a plethora of additional analyses that one could envisage, like separate analyses of each nuclear marker for improved species delimitation (but for this more samples of each lineage would need to be sequenced), more explicit model testing in terms of statistical phylogeography, etc – but most of these make little sense with the data set at hand and would only serve to unnecessarily inflate the paper. Overall, this is a great study absolutely meriting publication, and I look forward to seeing it published in RSOS.

Some of my main concerns relate to the summary where some aspects are presented too simplistic while in the main text they are usually discussed with the adequate care.

First sentence of Summary, and first sentence of Introduction: Be gentle with invasive species (of geckos)! Yes it is true that invasive species can become a severe threat! This is particularly true for introduced mammals predatory islands, or highly competitive weeds on islands, and sometimes for toxic species such as the cane toad. However, the vast majority of invasive species do no harm and sometimes even can enrich ecosystems (see the quite enlightening book of Fred Pearce " The New Wild: Why Invasive Species Will Be Nature's Salvation" – although there are good reasons to disagree with many of his statements).

But most importantly here, are there examples of invasive geckos that would constitute a true problem anywhere? I think there are indeed some problems with introduced *Phelsumas* on the Mascarenes, and maybe there are some other examples which I expect will always be in fragile island ecosystems. This should be explained in a more differentiated way – like it is now, a naive reader could think, invasive geckos are one of the biggest threats to global biodiversity, while certainly, the biggest threat continues being habitat destruction!

Response: We have rephrased the introduction to reflect the fact that invasive geckos need not impact native biota and added references: "Geckos include some of the most invasive species of reptiles, and in some cases there are known or predicted negative consequences of the invasives on native geckos (e.g. Hoskin 2011; Buckland et al. 2014; but see Olmedo and Cayot 1994). The cosmopolitan genus *Hemidactylus* stands out among geckos — ..."

line 21 and 24: please be consistent and here also use "putative species-level lineages". You use this almost everywhere else in the manuscript, but please double check. This is an important distinction – you are here not presenting any fully convincing, integrative evidence for any of these putative species, although of course I agree they probably all are distinct at the species level.

Response: We have made edits on lines 21, 24, 261 to reflect that these are putative species

line 23 (also line 260): also here, allow for uncertainty! You cannot state that these lineages originated in the Miocene ... but of course you could write that your analyses suggest such an age, or that you hypothesize this based on your analysis, or similar.

Response: we have edited both these, first adding "our analyses suggest" ...and on line 260 saying "likely"

421-422: Another quite strong statement! Yes, you have sequenced many specimens, but not enough for such a strong statement. Just add a "may" or change the phrasing otherwise!

Response: Our current phrasing states that our sampling suggests this, and we have added putative species here.

399: section Opportunity, ecological tolerance or a bit of both?

I would suggest two additional factor here (maybe to be listed in lines 444-451 or elsewhere in the section): behaviour and genetic variation! I am convinced that for many cases of lizards, and I suspect for many other invasive animals, aggressiveness and "dominant" behaviour are extremely important for their success as competitors. This in turn might be related to (not only intraspecific, but also intraindividual) genetic variation, a factor that might well be determinant to the adaptational capacity of an invasive propagule. The beauty of these two factors is that they could be easily tested, by behavioural experiments comparing different *Hemidactylus* species, and by genomic studies. I am pretty convinced successful invaders will on average turn out to be rather aggressive and highly active species, of course also with short generation times, and with high genetic variation mediating adaptability.

Response: We agree that these are important factors, and have added some text and references to reflect this : "Invasive geckos are known to displace native species based on (larger) body size, faster locomotion, and other traits linked to aggressive or dominant behaviour (e.g. Case and Bolger 1994;

Petren and Case 1996; Hoskin 2011; Short and Petren 2011). Understanding the factors that allow *H. mabouia* ss...". Our approach is simply to establish the patterns in this group and this paper sets the stage for such experiments.

Caption to Fig. 1: Please explain on what data the timetree is based. I suppose the concatenated nuclear + mitochondrial data?

Response: Added as suggested.